# Cognitive Function during the Prodromal Stage of Alzheimer’s Disease in Down Syndrome: Comparing Models

**DOI:** 10.3390/brainsci11091220

**Published:** 2021-09-16

**Authors:** Christy L. Hom, Katharine A. Kirby, Joni Ricks-Oddie, David B. Keator, Sharon J. Krinsky-McHale, Margaret B. Pulsifer, Herminia Diana Rosas, Florence Lai, Nicole Schupf, Ira T. Lott, Wayne Silverman

**Affiliations:** 1Department of Psychiatry and Human Behavior, University of California, Orange, CA 92868, USA; dbkeator@uci.edu; 2Center for Statistical Consulting, University of California, Irvine, CA 92697, USA; kathark@uci.edu (K.A.K.); jricksod@uci.edu (J.R.-O.); 3Institute for Clinical and Translational Sciences, University of California, Irvine, CA 92697, USA; 4New York State Institute for Basic Research in Developmental Disabilities, Staten Island, NY 10314, USA; sharon.krinsky-mchale@opwdd.ny.gov; 5Department of Psychiatry, Massachusetts General Hospital, Harvard Medical School, Boston, MA 02114, USA; mpulsifer@mgh.harvard.edu; 6Department of Neurology, Massachusetts General Hospital, Harvard Medical School, Boston, MA 02114, USA; rosas@helix.mgh.harvard.edu (H.D.R.); flai@partners.org (F.L.); 7Department of Neurology, College of Physicians and Surgeons, Columbia University, New York, NY 10027, USA; ns24@cumc.columbia.edu; 8Department of Epidemiology, School of Public Health, Columbia University, New York, NY 10027, USA; 9Department of Pediatrics, University of California, Orange, CA 92868, USA; itlott@uci.edu (I.T.L.); wsilverm@uci.edu (W.S.)

**Keywords:** mild cognitive impairment, Alzheimer’s disease, Down syndrome, cognitive decline, neuropsychological tests, cognitive function

## Abstract

Accurate identification of the prodromal stage of Alzheimer’s disease (AD), known as mild cognitive impairment (MCI), in adults with Down syndrome (MCI-DS) has been challenging because there are no established diagnostic criteria that can be applied for people with lifelong intellectual disabilities (ID). As such, the sequence of cognitive decline in adults with DS has been difficult to ascertain, and it is possible that domain constructs characterizing cognitive function in neurotypical adults do not generalize to this high-risk population. The present study examined associations among multiple measures of cognitive function in adults with DS, either prior to or during the prodromal stage of AD to determine, through multiple statistical techniques, the measures that reflected the same underlying domains of processing. Participants included 144 adults with DS 40–82 years of age, all enrolled in a larger, multidisciplinary study examining biomarkers of AD in adults with DS. All participants had mild or moderate lifelong intellectual disabilities. Overall AD-related clinical status was rated for each individual during a personalized consensus conference that considered performance as well as health status, with 103 participants considered cognitively stable (CS) and 41 to have MCI-DS. Analyses of 17 variables derived from 10 tests of cognition indicated that performance reflected three underlying factors: language/executive function, memory, and visuomotor. All three domain composite scores significantly predicted MCI-DS status. Based upon path modeling, the language/executive function composite score was the most affected by prodromal AD. However, based upon structural equation modeling, tests assessing the latent construct of memory were the most impacted, followed by those assessing visuomotor, and then those assessing language/executive function. Our study provides clear evidence that cognitive functioning in older adults with DS can be characterized at the cognitive domain level, but the statistical methods selected and the inclusion or exclusion of certain covariates may lead to different conclusions. Best practice requires investigators to understand the internal structure of their variables and to provide evidence that their variables assess their intended constructs.

## 1. Introduction

Early detection of Alzheimer’s disease (AD) is of paramount importance to enhance efficacy of clinical intervention and to improve understanding of AD progression in people with Down syndrome (DS). Due to the triplication of the amyloid precursor protein gene on chromosome 21, people with DS have life-long overproduction of amyloid-β, and the earlier production of amyloid- β in this population, detectable in imaging studies as early as 14 years of age, contributes to increased AD risk [1,2]. That is, adults with DS are likely to experience earlier AD-related cognitive declines than their counterparts in the neurotypical population [3].

Complicating early detection of AD progression and diagnostic accuracy is the large variability in baseline cognitive ability among people with DS, ranging from borderline intellectual functioning to profound impairment [2,4,5]. Diagnosis of any form of intellectual disability (ID) requires an intelligence quotient (IQ) that is two or more standard deviations (SD) below the population mean, and the range in IQ within the population with DS can be more than 50 points [6], equivalent to over 3 SDs. In the neurotypical population, the prodromal stage of dementia (i.e., mild cognitive impairment; MCI) is typically diagnosed based upon decline, together with performance that is 1.5 SDs or more below the population mean on cognitive tests [7,8]). However, this criterion cannot be applied to the population with DS since the vast majority of affected individuals have functioned more than 1.5 SDs below the mean since childhood [2,3]. The lack of data on normative age-related declines in cognition among healthy adults with DS also makes it difficult to differentiate between expected age-related declines and MCI [9]. A 50-year study by Carr and Collins [10] is one of the few longitudinal investigations in people with DS that has been able to isolate age-related from disease-related changes, and they found that in the absence of dementia, people with DS experienced significant age-related changes in some cognitive domains (e.g., memory and non-verbal IQ) but not others (e.g., receptive and expressive language skills).

The lack of agreed-upon diagnostic criteria for dementia specific to people with ID, in the literature or in practice, further complicates the diagnostic process [11], which is why Basten et al. [12] argued that recognition and treatment of the neurodegenerative components of the syndrome are the greatest unmet therapeutic needs. The lack of specifics regarding the subtle cognitive changes that characterize prodromal AD not only affects accuracy of diagnosis and the timing of pharmacological interventions for the population with DS, but it also contributes to their being overlooked for inclusion in AD randomized controlled trials [13].

Numerous investigations have been undertaken to characterize the cognitive changes associated with AD in persons with DS. Most of these investigations have been unsuccessful in identifying consistent or clear differences between those in the preclinical and prodromal stages of AD [13,14]. Some found changes in episodic memory to be prominent during the prodromal and early stage of AD in DS [15], whereas others found changes in personality/behavior and executive function to be more prominent than deterioration in episodic memory in persons with severe to mild ID [16,17,18,19,20,21]. Still others found reduced language skills [22,23,24] and adaptive functioning [25,26] to be the earliest indicators. After a systematic review of the DS-AD literature, Lautarescu et al. [27] concluded that some of the variability in presentation during the early stages of AD in the population with DS may be due to differences in premorbid intellectual capacity and each individual’s ability to compensate for newly acquired deficits.

The present study explores whether cognitive deficits characterizing the prodromal stage of AD in individuals with DS reflect the same underlying domains when severity of intellectual disability ranges from “moderate” to “mild”. Specifically, we examine tasks requiring executive function, language, memory, or visuomotor coordination, areas previously found to be particularly affected by dementia in adults with DS [28]. We also examine age-related declines in this sample that is at high risk for AD neuropathology as well as compare different methods for modeling the cognitive decline associated with AD progression. Due to the exploratory nature of these analyses, we did not specify a priori a hierarchy of deterioration among cognitive domains.

## 2. Materials and Methods

### 2.1. Participants and Procedure

All procedures were reviewed and approved by the Institutional Review Boards at participating institutions (New York State Institute for Basic Research in Developmental Disabilities, Columbia University Irving Medical Center, Massachusetts General Hospital, the University of California at Irvine, The Johns Hopkins University Schools of Medicine and Public Health, and the University of North Texas Health Science Center). Informed consent was obtained from participants or their legally authorized representatives along with participant assent. The current analyses included data from 144 adults with DS, ranging from 40 to 81 years of age, who were enrolled in a larger, multidisciplinary program of research focused on biomarkers of AD in adults with DS. Only the participants assessed in New York, New York (*n* = 43); Boston, Massachusetts (*n* = 56); and Irvine, California (*n* = 46) were administered the full neuropsychological battery that is reported in this study. Therefore, participants assessed at other sites could not be included in the current set of analyses. Inclusion criteria included (1) age ≥ 40, (2) estimated preexisting IQ > 30, (3) absence of significant sensory or motor impairments, and (4) willingness to provide a routine blood sample for studies of fluid-based biomarkers of AD.

Participants received a comprehensive evaluation at study baseline that included (a) a review of medical records; (b) physical and neurological evaluations; (c) interviews with knowledgeable informants focused on cognitive and functional abilities, health-related condition and medical history, and neuropsychiatric concerns; and (d) direct one-on-one testing with a core battery developed specifically for assessing dementia status in adults with intellectual disabilities. The full direct testing battery required approximately 1.5 to 2 h to complete. None of the participants in the larger, multidisciplinary program of research with severe ID had valid scores on all 22 measures; therefore, this subgroup was not included in the current study.

### 2.2. Consensus Disease Status

Following the comprehensive evaluation, each participant’s AD-related disease status was determined through a Consensus Review Conference (see [29]) that included program investigators at the respective enrollment site, senior staff members, and research staff who had direct contact with the participant under consideration. Disease status was classified into the following categories: (a) Cognitively Stable (CS), indicating with reasonable certainty that AD-related impairment was absent (although allowing for declines normally expected to accompany aging, per se); (b) MCI-DS, indicating that there was some indication of cognitive and/or functional decline beyond what would be expected with aging, per se, but of insufficient severity to suggest frank dementia; (c) Possible Dementia, indicating that some signs and symptoms of dementia were present but were not judged to be totally convincing; (d) Definite Dementia, indicating with high confidence that dementia was present; and (e) Uncertain (due to complications), indicating that evidence of clinically significant declines were present but might be caused by some other substantial concern, usually a medical condition unrelated to a dementing disorder or a significant life event (e.g., severe sensory loss, poorly resolved hip fracture, death of a loved one). For the current study, only data from the CS and MCI-DS groups were analyzed.

### 2.3. Measures

The core neuropsychological battery consisted of instruments previously demonstrated to be valid for use with adults having DS and that covered the spectrum of cognitive domains expected to be affected by clinical progression of AD, including during its prodromal stage. Twenty-two measures from the following 13 instruments were hypothesized a priori to measure four cognitive domains: (1) executive function, (2) language, (3) memory, and (4) visuomotor:The Beery Buktenica Developmental Test of Visual-Motor Integration—long form (VMI; [30]) assesses visual-motor integration skills. The total raw score is used.The Block Design subtest from the Wechsler Intelligence Scale for Children, 4th Edition [31] supplemented with less complex items from the original Down Syndrome Mental Status Examination (DSMSE) [32] assesses visual–spatial reasoning and visual–motor dexterity. The total raw score is used.The Boston Naming Test [33] measures confrontational picture-naming abilities. The measure of performance is total correct, with or without a semantic cue.An adaptation of the Category Fluency Test [34] with slightly liberalized scoring measures semantic fluency. Participants are asked to generate as many words as they can within one of three categories (food, animals, clothing) in 20 s. The measure of performance is the total number of words generated in two categories, excluding repetitions.The Cats and Dogs Task [20] assesses response inhibition using a Stroop test paradigm [35]. The measure of performance is the amount of time used to name all of the animals as printed on the sheet (Naming condition) subtracted from the amount of time used to state the opposite animal (Switch condition). The number of errors on the Naming and Switch conditions are also recorded.The Cued Recall Test [36] assesses verbal learning and memory. The measure of performance is total number correct across three test trials.An enhanced version of the DSMSE [32] that expands the number of items included in tests of short-term memory (from 3 to 9 objects) assesses several different abilities. Three subscale scores are used: Language, Memory, and Visual Spatial.A simplified version of the Modified Mini Mental Status Evaluation (mMMSE) [37,38] assesses several different abilities. Four subscales are used: Anomia, Concentration, Fine Motor, and combined Orientation (Person, Place, and Time).The Rapid Assessment of Developmental Disabilities, Second Edition (RADD-2) [39] is a battery of items from commonly used tests, including the Wechsler Intelligence Scale for Children, 4th Edition; Hawaii Early Learning Profile [40]; and Merrill-Palmer-Revised Scales of Early Childhood [41]. The Digit Span Forward, Expressive Language, Hand Movements, Imitation, Receptive Language, and Similarities subscales are used. Since the Hand Movements and Imitations subscales both assess motor coordination, they were combined into one score for our analyses, labeled “sensorimotor”.Purdue Pegboard [42] assesses hand dexterity. The measure of performance is the number of pairs completed with both hands simultaneously within one minute.The Rivermead Behavioural Memory Test [43] assesses visual memory and recognition. The measures of performance are the number correct during the identification trial and the number correct minus the number incorrect during the recognition trial.A modified version of the Selective Reminding Test (SRT), now requiring free recall of 8 items over 3 trials (see [15,44]), assesses verbal learning and memory. The measure of performance is total correct over 3 trials.The Tinetti Balance and Gait Assessment Tool [45] assesses gait and balance on a three-point ordinal scale with a range of 0 to 2. A score of 0 represents the most impairment, whereas a score of 2 represents independence. We only administered the Gait tool, and that score is presented.

### 2.4. Statistical Analyses

Statistical analyses were performed using Stata version 16. Age differences between the CS and MCI-DS groups were evaluated through a Mann–Whitney test because age was not normally distributed. All other demographic and health comparisons were evaluated through Pearson chi-square tests. Before evaluating how well our measures predicted MCI-DS status, we examined the underlying factor structure of all the measures, stratified by premorbid ID level, using an exploratory factor analysis (EFA) with oblique promax rotation to take account of the correlated nature of the domains and increase the interpretability of the factor pattern matrix. Age-related cognitive deficits were examined by performing separate linear regression models for each test, first for the CS group, then for the MCI-DS group. Domain composite scores were created by using the sum of all rescaled test scores that were hypothesized to assess that single domain. Test scores were rescaled using the Proportion of Maximum Scoring [46] (POMS), which uses a 0–1 range to show the magnitudes of associations among variables without changing the shape of the distribution. Path modeling was used to assess the sensitivity of domain composites in identifying MCI-DS, adjusting for sex and premorbid ID. Structural equation modeling (SEM) was used to assess the association between MCI-DS and each latent variable, adjusting for sex and premorbid ID. Domain labels were based upon the results of EFA and an a priori factor structure. Model fit of the path and final SEM models were compared using multiple fit indices, including the chi-square goodness of fit, comparative fit index (CFI), Tucker–Lewis Index (TLI), and root mean square error of approximation (RMSEA). An RMSEA below 0.08 and a CFI and TLI above 0.90 are considered to indicate acceptable fit [47,48].

## 3. Results

### 3.1. Demographics

In the full sample, 59.0% were male, 53.5% had premorbid ID in the mild range, and 46.5% in the moderate ID range. Most participants were white (86.1%), followed by Hispanic (6.9%), Asian (4.2%), Black (2.1%), and American Indian (0.7%). In terms of chromosomal diagnosis, 84.0% had full trisomy 21, 4.9% were mosaic, 3.5% had translocation DS, and 7.7% were unknown. Regarding AD status, 71.5% were CS and 28.5% were MCI-DS. The MCI-DS group was significantly older, z = −3.39, *p* = 0.0001, and were reported to have more co-occurring health problems than the CS group, but the rates of chronic medical conditions did not differ significantly (*p*s = 0.126 to 0.894, see Table 1).

### 3.2. Test Outcomes and Cognitive Domains

Table 2 presents the means and standard deviations for each test and their hypothesized cognitive domain. Outcomes on all measures were in the expected direction (CS > MCI-DS).

### 3.3. Underlying Structure

A series of exploratory factor analyses (EFAs) were performed to examine how well the cognitive variables loaded onto the four hypothesized domains. An initial set of analyses considered 22 scores, but results revealed problems with the following five variables: (1) the Tinetti Gait measure did not load on the first four factors and had a uniqueness of 0.93, meaning that much of the information contained in the variable was not predicted by any factor; (2) Cats and Dogs performance time did not load on the first four factors and had a uniqueness of 0.96, whereas the error score loaded on the factor describing memory and 28% of the sample had less than 50% accuracy; (3) mMMSE Orientation and mMMSE Concentration loaded on the same factor in every iteration of EFA, so the Orientation measure was dropped due to consistently lower loadings than the Concentration measure and a narrower range of abilities tested; (4) mMMSE Fine Motor loaded onto the factor that assessed language and executive function instead of visuomotor because it required participants to know how to write numbers and letters in order; and (5) Category Fluency loaded onto multiple factors and the loadings were weaker in relation to the other tests within the same factor (0.32 to 0.36). These five variables were therefore dropped from further analyses.

The results of the EFA with promax rotation for the 17 remaining variables are summarized in Table 3. We retained three factors since the third factor had an eigenvalue close to 1 (0.95). EFAs were also performed separately for the two levels of premorbid ID. Both analyses yielded similar results, with three factors accounting for 91.65% of the total variance in the mild ID group and 90.75% of the total variance in the moderate ID group. Furthermore, each measure loaded onto the same factor, regardless of premorbid level of ID. The three factors consistently identified by each EFA were (1) language/executive function, (2) memory, and (3) visuomotor, confirming three out of four a priori hypothesized domains. These three factors explained 97.55% of the total variance in scores among the full sample across all levels of ID.

Although our executive function measures all required language skills to a certain extent, we did not expect them to be as highly correlated with the language measures (r^2^ = 0.53 to 0.72, *p* < 0.001) as they were with each other (r^2^ = 0.67, *p* < 0.001). As a result, tests from both domains were combined into a single factor, labeled “language/executive function.” As mentioned above, two of the other tests that were hypothesized to measure executive function (Category Fluency and Cats and Dogs) were dropped from all additional analyses because they lacked clear association with a single factor that had an eigenvalue of ≥1 and/or had relatively weak loadings (defined a priori as anything < 0.30).

### 3.4. Effects of Aging

Linear regression models were performed to evaluate the effects of aging independent of disease status. Scatterplots and the fitted regression line for each cognitive test are presented as Appendix A. There was a significant main effect for aging on nine of the 17 measures, controlling for disease status. The three memory and two language measures that were affected by aging were DSMSE Language, DSMSE Memory, Purdue Pegboard, RADD-2 Expressive Language, RADD-2 Receptive Language, Rivermead Recognition, and SRT (Fs(3, 130) = 3.33 to 25.48, ts = −3.62 to −2.01, *p*s = 0.02 to 0.0001). Since there was only one significant interaction between age and disease status, RADD-2 Sensorimotor (F(3, 130) = 5.04, t = −5.08, *p* = 0.002), we decided not to include age as a covariate in all subsequent analyses in order to minimize the likelihood of a type II error.

### 3.5. Path Model

Path analysis was used to examine the magnitude and significance of the three domain composites, adjusting for sex and premorbid ID level. Every domain composite was significantly related to MCI-DS status (see Table 4, but the path model only explained about half of the variance between CS and MCI-DS (R^2^ = 0.49) and had poor model fit, as the RMSEA was 0.49 (much greater than the 0.08 limit), CFI was 0.51 (not close to the 0.90 threshold), and TLI was −0.94 (far from the 0.90 cutoff).

MCI-DS status was associated with lower scores on every domain composite, with the language/EF composite being the most affected, followed by the memory composite, then the visuomotor composite.

### 3.6. Structural Equation Model

Figure 1 displays the relationship between the latent, measurement, and exogenous variables (sex, premorbid ID, and MCI-DS status, respectively). The 17 cognitive measures and their latent variables predicted MCI-DS status substantially better than the composite scores used in the path model. The SEM model had good model fitness since RMSEA = 0.08, CFI = 0.91, and TLI = 0.90.

Sex was not related to outcomes on any of the cognitive measures. MCI-DS status was associated with greater memory impairment than premorbid ID level was, whereas premorbid ID level was associated with greater language/EF impairment than MCI-DS status was. MCI-DS status and premorbid ID had similar effects on visuomotor performance.

As shown in Table 5, the latent construct of memory was a stronger indicator of MCI-DS status than the latent construct of visuomotor, which was a stronger indictor than language/EF. Although the path and SEM models indicated the same directionality of domain impairments for those with MCI-DS, the magnitude and ranking were different. (The path model identified the language/EF composite to be a stronger indicator of MCI-DS than the memory and visuomotor composites.) In short, giving all tests within the same domain differential weights as in the SEM can lead to considerably different conclusions than when all tests within the same domain are given equal weight as in the path analysis.

The largest language/EF differences between the CS and MCI-DS groups were observed on the following tests: Boston Naming, DSMSE Language, RADD-2 Expressive Language, RADD-2 Digit Span Forward, and RADD-2 Similarities (ẞs = 0.76 to 0.88). The largest differences between the two groups on memory tests were on DSMSE Memory and SRT (ẞs = 0.85 to 0.86). The largest differences between the two groups on visuomotor tests were on Block Design and DSMSE Visual Spatial (ẞs = 0.80 to 0.85).

## 4. Discussion

The population with DS is unique from other populations at risk for AD due to atypical brain development, lifelong amyloid overproduction, and significant but variable cognitive impairments present prior to clinical manifestations of AD. The heterogeneity in premorbid functioning, lack of clear diagnostic criteria for this population, and use of cognitive tests that were designed for neurotypical populations have contributed to some of the difficulties in characterizing their cognitive changes during the prodromal stage of AD [19,49]. Our study of 144 older adults with DS confirmed that our extensive neuropsychological battery indeed measured cognitive functioning in at least 3 domains: language/executive functioning, memory, and visuomotor. These domains are widely investigated amongst neurotypical populations, but few studies have demonstrated that the cognitive tests used in research for people with DS measure such functions in older adults experiencing AD-related neuropathology.

Accordingly, it is important to highlight that a few of the tests in our battery that were expected to measure executive function had to be dropped from our models because of high error rates, likely associated with low comprehension of task demands. Furthermore, the tasks that were retained failed to show predicted distinctions between language and executive function skills. This underscores the need to identify or develop other tasks of executive function for this population and to evaluate their relationship(s) with tasks targeting different underlying processes—for example, Category Fluency, loaded on the language and visuomotor domains instead of with the other executive function measures, and mMMSE Fine Motor, loaded on the memory domain instead of the visuomotor domain because the task required recall of numbers and letters in the correct order. These are important reminders that cognition is multifaceted and related to multiple brain networks. Few cognitive tests solely measure one specific cognitive skill [50], and to some extent, successful performance on any test requires a certain amount of motivation, attention, comprehension, and memory. For the purposes of this paper, we only selected measures that loaded on a single domain to facilitate the interpretation of results.

### 4.1. Age-Related Impairments

Since aging is closely related to AD progression, we wanted to identify test outcomes that were most affected by the normal aging process and to examine whether age-related impairments differed by disease status. Aging significantly impacted performance on more than half of the cognitive tests (9 of 17) and in every domain. However, there was only one significant age x disease status interaction (RADD-2 Sensorimotor), indicating that AD progression may accelerate age-related declines on visuomotor tasks. That is, age may be a moderator, but it is not a confounder.

### 4.2. Premorbid Functioning

Differences in premorbid intellectual capacity did not alter the factor structure of our neuropsychological battery, but it did affect performance on all the cognitive measures. As expected, the group with mild premorbid ID consistently scored higher than the group with moderate premorbid ID. This emphasizes the importance of taking premorbid functioning into consideration in all analyses for this population due to such heterogeneity in baseline cognitive abilities. The use of a universal criterion, such as 1.5 SD below the population mean, is not appropriate for people with DS, even if the population mean is derived from other individuals with DS, because baseline IQ scores in this population can range from 2 to 4 SDs below normative levels (i.e., borderline functioning to profound ID). Nevertheless, our results provide evidence that cognitive functioning can be characterized by the same cognitive tests and domains in this population when their premorbid impairment is in the mild to moderate range.

### 4.3. Composites vs. Individual Test Scores

All three domain composites, which weighted all tests equally, were significantly related to MCI-DS status, but they only explained half of the total variance in test scores. In contrast, individual test scores and their regression weights explained most of the variance and had good model fit. Therefore, the cognitive features of prodromal AD in DS may be better ascertained by using re-scaled individual test scores rather than by using a domain average or sum. Additionally, the domain that was identified as being the most impacted by prodromal AD using composite scores was different than the one identified using regression weights. Composite scores indicated language/executive function skills to be the most affected, whereas individual test scores indicated the latent variable of memory to be the most affected. This accentuates the strength of estimating each domain using a factor analysis as opposed to a priori sum or mean scores.

This study demonstrates the possibility of drawing different conclusions from the same sample and outcome measures based upon the type of statistical analyses used and the factors selected for inclusion as covariates. Differences in analytical approaches may explain the conflicting findings reported between several large-scale studies on adults with DS since the neuropathological cascade caused by underlying AD should be similar for all persons with DS. For example, Cosgrave et al. [51] and Krinsky-McHale et al. [15] found memory decline to be the earliest sign of AD, whereas Ball et al. [20], Adams and Oliver [52], and Fonseca et al. [49] found executive function decline (e.g., planning, inhibitory control, working memory, and abstract thinking) to be the earlier indicator of AD progression. Still others have found language decline to be more indicative of early AD [53]. Future studies should explore the psychometric properties of their variables before deciding whether to use multiple scales or one unidimensional sum score. As McNeish and Wolf [54] illustrated, sum scoring can lead to different conclusions compared to more rigorous methods of factor estimation, and multilevel models, growth models, or multiple regression based on sum scores may be adversely affected by imprecision when summing multiple scales.

Finally, our sample of adults with DS in the pre-clinical stages of AD offers normative data on a broad array of cognitive tests that have been used among older adults in this population. It is our hope that this will assist future investigations in identifying the effects of pathological and normative aging in this population.

### 4.4. Limitations

Despite our best attempts to maximize participants’ motivation and attention during 1.5 h of cognitive testing, including the option to administer the battery over the course of two separate visits and flexibility in the number of breaks given, we cannot rule out the possibility that test performance was influenced by these factors or language comprehension skills. Another caveat is that alternative procedures for measuring a hypothesized skill could tap multiple domains or other domains that were not included in our analyses. Hence, our factor solution may not apply to other cognitive batteries. Studies that use a different combination of tests should examine the underlying structure of their battery rather than rely upon our cognitive domain classifications. Another limitation of our study is that classification of disease status, although conducted by teams with extensive experience working with this population, is an inherently imperfect process (as it is for diagnosis of prodromal AD in elderly adults with a neurotypical developmental history). Moreover, there was some circularity in the use of certain measures of the cognitive battery to aid in consensus diagnoses, which we believe was minimized by including the results of a physical exam and a variety of informant-based measures.

Lastly, none of the participants in our larger, multisite study with severe ID successfully completed all cognitive measures. Hence, the findings in the current study should not be generalized to those individuals, a significant minority of adults with DS. This points to the need to develop and evaluate procedures specifically targeting AD clinical progression in adults with more severe lifelong disabilities.

## 5. Conclusions

The cognitive abilities of individuals with atypical brain development, early onset of AD pathology, and accelerated rates of amyloid accumulation may be characterized by the same cognitive domains described for neurotypical populations. However, the complex pathobiology of DS leads to both physical deficits and biochemical changes that can lead to multiple comorbid medical conditions, genetic and epigenetic variation, environmental factors, and stochastic events [14], all contributing to considerable heterogeneity in this population. Just as there are multiple cognitive phenotypes of MCI in the neurotypical population, there may also be more than one cognitive phenotype of MCI-DS.

## Figures and Tables

**Figure 1 brainsci-11-01220-f001:**
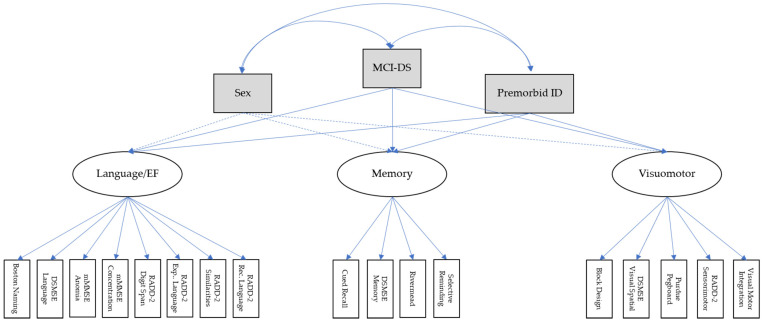
Structural equation model of how MCI-DS affected performance on the 17 cognitive measures and 3 latent variables. Solid lines indicate paths that were statistically significant, *p*s = 0.001. Ovals represent the endogenous variables with their indicator variables represented by white rectangles. CFI = Comparative Fit Index, CI = confidence interval, DSMSE = Down Syndrome Mental Status Examination, EF = executive function, Exp = expressive, ID = intellectual disability, mMMSE = Modified Mini-Mental Status Exam, MCI-DS = Mild Cognitive Impairment-Down syndrome, RADD-2 = Rapid Assessment of Developmental Disabilities-2nd Edition, Rec = receptive, RMSEA = root mean square error of approximation, Model Fit Indices: Chi-square = 299.20, CFI = 0.91, RMSEA = 0.08, 90% CI = 0.07–0.09.

**Table 1 brainsci-11-01220-t001:** Demographic and health comorbidities by AD status.

Condition	CS (n = 103)	MCI-DS (n = 41)	Mann–Whitney/χ^2^ Statistic	*p*-Value
Age	M = 48.65, SD = 6.27	M = 52.88, SD = 6.72	−3.39	0.0001
Sex	Male (56.31%)	Male (65.85%)	1.10	0.293
Premorbid ID	Mild (58.25%) Moderate (41.75%)	Mild (41.46%) Moderate (58.54%)	3.32	0.068
Depression	28.16%	29.27%	0.02	0.894
Diabetes	6.80%	9.76%	0.75	0.688
Hearing	Corrected (15.53%) Impaired (16.50%)	Corrected (19.51%) Impaired (19.51%)	0.65	0.723
Hypertension	7.46%	11.76%	0.51	0.473
Obstructive sleep apnea	30.10%	41.46%	4.14	0.126
Seizures	11.65%	9.76%	0.94	0.625
Vision	Corrected (56.41%) Impaired (17.48%)	Corrected (56.10%) Impaired (29.27%)	3.67	0.159

**Table 2 brainsci-11-01220-t002:** Means and standard deviations of raw test scores by AD status and their hypothesized cognitive domain.

Variable (Range of Scores)	Domain	CS (*n* = 103)	MCI-DS (*n* = 41)	Mann–Whitney U (*p*-Value)
Block Design (0–54)	Visuomotor	23.79 (10.49)	17.37 (12.78)	3.11 (0.002)
Boston Naming (0–27)	Language	15.88 (5.51)	13.23 (6.91)	2.04 (0.041)
Category Fluency (0–17)	EF	8.23 (3.18)	6.82 (3.68)	2.21 (0.027)
Cats and Dogs Switch (−17.00–61.80) ^†^	EF	9.80 (11.22)	5.05 (11.21)	1.82 (0.069)
Cued Recall (3–35)	Memory	28.61 (6.83)	21.38 (9.22)	4.43 (0.0001)
DSMSE Language (3–52)	Language	37.10 (9.07)	30.80 (9.63)	3.28 (0.001)
DSMSE Memory (0–23)	Memory	14.08 (4.68)	9.80 (4.57)	4.62 (0.0001)
DSMSE Visual Spatial (2–8)	Visuomotor	6.18 (1.09)	5.59 (1.01)	2.98 (0.003)
mMMSE-DS Anomia (4–20)	Language	18.18 (2.20)	16.49 (4.19)	1.55 (0.122)
mMMSE-DS Concentration (0–6)	EF	3.69 (2.10)	2.39 (2.14)	3.15 (0.002)
mMMSE-DS Fine Motor (1–10)	Visuomotor	8.04 (1.29)	7.05 (2.28)	2.58 (0.010)
mMMSE-DS Orientation (5–30)	EF	25.46 (5.33)	20.87 (6.41)	4.75 (0.0001)
Purdue Pegboard Both Hands (0–8)	Visuomotor	2.62 (1.82)	1.51 (1.71)	3.14 (0.002)
RADD-2 Digit Span Forward (0–8)	EF	4.02 (1.66)	2.97 (1.80)	2.84 (0.005)
RADD-2 Expressive Lang. (0–16)	Language	11.24 (3.92)	8.80 (4.01)	3.34 (0.001)
RADD-2 Receptive Lang. (0–12)	Language	7.56 (2.51)	6.80 (2.99)	1.08 (0.279)
RADD-2 Sensorimotor (0–7)	Visuomotor	6.70 (0.50)	5.85 (1.50)	3.87 (0.0001)
RADD-2 Similarities (0–4)	Language	2.51 (1.49)	1.56 (1.47)	3.40 (0.001)
Rivermead Recognition (0–10)	Memory	5.42 (3.77)	2.21 (3.40)	3.90 (0.0001)
Selective Reminding Test (1–24)	Memory	15.50 (5.62)	9.18 (4.32)	5.70 (0.0001)
Tinetti Gait (4–12)	Visuomotor	10.75 (1.62)	10.37 (1.88)	1.47 (0.143)
VMI (1–25)	Visuomotor	15.45 (3.22)	13.98 (3.40)	1.75 (0.081)

^†^ Cats and Dogs is measured in seconds; all other scores are measured in points (number correct). EF = executive function.

**Table 3 brainsci-11-01220-t003:** Factor structure of the 17 retained cognitive variables across all levels of ID (*n* = 128).

	1	2	3
	Language/Executive Function	Visuomotor	Memory
EXECUTIVE FUNCTION			
mMMSE Concentration	0.67		
RADD-2 Digit Span Forward	0.82		
LANGUAGE			
Boston Naming	0.91		
DSMSE Language	0.98		
mMMSE Anomia	0.70		
RADD-2 Expressive Language	0.79		
RADD-2 Receptive Language	0.61		
RADD-2 Similarities	0.80		
MEMORY			
Cued Recall			0.80
DSMSE Memory			0.80
Rivermead Recognition			0.62
Selective Reminding Test			0.66
VISUOMOTOR			
Block Design		0.80	
DSMSE Visual Spatial		0.83	
Purdue Pegboard		0.49	
RADD-2 Sensorimotor		0.57	
VMI		0.54	
Percentage of variance	74.13	16.51	6.91

Blanks represent absolute loading < 0.30.

**Table 4 brainsci-11-01220-t004:** Path model comparing CS to MCI-DS adjusted for sex and premorbid ID level (*n* = 144).

					95% CI	
Domain Score	β	SE	Z	P	Lower	Upper
Language/Executive Function	−0.97	0.29	−3.34	0.001	1.54	−0.40
Memory	−0.89	0.15	−6.04	0.0001	−1.17	−0.60
Visuomotor	−0.53	0.12	−4.23	0.0001	−0.77	−0.28

**Table 5 brainsci-11-01220-t005:** The relationship between sex, premorbid ID, MCI-DS, and cognitive functioning.

Cognitive Domain	Predictor Variable	Estimate	SE	CR	Standardized Regression Weight	*p*-Value
Structural						
Language/EF	Sex	−0.010	0.075	−0.13	−0.01	0.893
Language/EF	PID	−0.413	0.067	−6.15	−0.41	<0.001
Language/EF	MCI-DS	−0.238	0.074	−3.23	−0.24	0.001
Memory	Sex	0.008	0.076	0.11	0.01	0.916
Memory	PID	−0.241	0.075	−3.21	−0.24	0.001
Memory	MCI-DS	−0.471	0.67	−7.07	−0.47	<0.001
Visuomotor	Sex	−0.093	0.081	−1.14	−0.09	0.255
Visuomotor	PID	−0.302	0.077	−3.91	−0.30	<0.001
Visuomotor	MCI-DS	−0.303	0.78	−3.90	−0.30	<0.001
Measurement						
Language/EF	Boston Naming	0.879	0.023	38.18	0.88	<0.001
Language/EF	DSMSE Language	0.878	0.023	37.81	0.88	<0.001
Language/EF	mMMSE Anomia	0.742	0.041	18.14	0.74	<0.001
Language/EF	mMMSE Concentration	0.724	0.043	16.53	0.72	<0.001
Language/EF	RADD-2 Digit Span Forward	0.782	0.036	21.93	0.78	<0.001
Language/EF	RADD-2 Expressive Language	0.874	0.023	37.52	0.87	<0.001
Language/EF	RADD-2 Receptive Language	0.683	0.047	16.53	0.68	<0.001
Language/EF	RADD-2 Similarities	0.760	0.038	20.05	0.76	<0.001
Memory	Cued Recall	0.673	0.053	12.65	0.67	<0.001
Memory	DSMSE Memory	0.858	0.032	27.09	0.86	<0.001
Memory	Rivermead Recognition	0.702	0.051	13.87	0.70	<0.001
Memory	Selective Reminding Test	0.845	0.033	25.65	0.85	<0.001
Visuomotor	Block Design	0.849	0.033	26.09	0.85	<0.001
Visuomotor	DSMSE Visual Spatial	0.797	0.038	17.05	0.80	<0.001
Visuomotor	Purdue Pegboard	0.615	0.59	10.39	0.62	<0.001
Visuomotor	RADD Sensorimotor	0.624	0.058	10.70	0.62	<0.001
Visuomotor	VMI	0.708	0.048	14.77	0.71	<0.001

CR = critical ratio, EF = executive function, MCI = mild cognitive impairment, PID = premorbid level of ID, SE = standard error.

## Data Availability

The data presented in this study are openly available in the LONI Image and Data Archive (IDA) repository at https://ida.loni.usc.edu/login.jsp?project=ABCDS (accessed on 28 October 2020).

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
