# Peer review of "Cognitive Function during the Prodromal Stage of Alzheimer’s Disease in Down Syndrome: Comparing Models"

_brainsci, 2021, doi:10.3390/brainsci11091220_

Round 1

Reviewer 1 Report

This is an exceptionally well written manuscript that clearly lays out the problem, the approach, and the results and discussion.  The question tackled is an important one - how to improve the reliable detection of MCI in Down Syndrome patients who have mild to moderate intellectual disability.  The approach taken is sound, the findings are impactful, and the interpretation is well reasoned and defensible.  The statistical models gave somewhat different solutions about the relative utility of different cognitive domains in discriminating between cases that are stable cognitively or show emerging disease.  The results have clinical implications for improving the diagnosis in Down Syndrome and also illustrate the relative utility of different statistical models (with and without differential weighting of tests in the same domain) for improving the detection of disease. There are applications here in this study that extend well beyond this condition, Down Syndrome, to other disorders involving broad batteries of cognitive tests.  The major limitations of the study have all been essentially identified and appropriately discussed. The only other limitation to consider is that different solutions are possible with different inputs: tests or procedures that tap these same domains or tap other domains that were not included.  This is a relatively minor point. There are no concerns.

Author Response

We agree with your suggestion about there being alternative possible solutions if different tests or procedures were used and have included this caveat in our Limitations section.

Reviewer 2 Report

This is an interesting manuscript. However, there are some concerns:

Methods & Results:

Lines 258 and 261 MMSE was misspelled as MMMSE.

The rationales for dropping the variables when conducting the EFA are not convincing, as some are not statistically driven but rather theoretically driven, defeating the purpose of conducting the analysis.

Block design loaded onto the memory instead of other domains is interesting and at the same time concerning.

Lines 283-284: Although the test domains were highly correlated, combining them may prove troublesome, as the EFA clearly detected them as two separate factors.

Why was path model chosen to compare CS to MCI-DS? A much simpler MANCOVA would do the job.

Figure 1. …Solid lines indicate paths that were statistically… it seems to be lacking a word here.

The differentiation between the domain composite scores and the factors scores of the neurocognitive tests were not clear. It is advisable to clearly differentiate the two (one is statistically-driven, the other was theoretically-driven) early on in the introduction and clearly in the statistical analyses sub-section. The naming conventions for the two in the manuscript are confusing as well.

One way to avoid the confusion on the aforementioned is to group sections 3.4 and 3.5 together, and sections 3.3 and 3.6 together.

Relatedly, why was the path model tested for the theoretically-driven composite but not the factor scores?

Lines 339-343: “the strongest” refer to the B coefficients, presumably. However, the strongest against which variable was not clear here.

Discussion:

Line 422: I would not say this is a large sample for this condition. Moderate would be more appropriate.

Round 2

Reviewer 2 Report

all comments have been addressed, thank you for the comprehensive responses.